# Health care workers intention to accept COVID-19 vaccine and associated factors in southwestern Ethiopia, 2021

**Abiy Tadesse Angelo** *, **Daniel Shiferaw Alemayehu, Aklilu Mamo Dachew**

Department of Nursing, Mizan-Tepi University, Mizan-Aman, Ethiopia

* abiyutad@gmail.com

## Abstract

### Introduction

Health care workers are the most affected part of the world population due to the COVID-19 pandemic. Countries prioritize vaccinating health workers against COVID-19 because of their susceptibility to the virus. However, the acceptability of the vaccine varies across populations. Thus, this study aimed to determine the health care worker's intentions to accept the COVID-19 vaccine and its associated factors in southwestern Ethiopia, 2021.

### Methods

A facility-based cross-sectional study was conducted among health care workers in public hospitals in southwestern Ethiopia from March 15 to 28, 2021. A simple random sampling method was used to select 405 participants from each hospital. Data were collected using self-administered questionnaires. Descriptive statistics, such as frequency and percentage, were calculated. Multivariable logistic regression was also performed to identify factors associated with health care worker's intention to accept the COVID-19 vaccine. Statistically significant variables were selected based on p-values (<0.05) and the adjusted odds ratio was used to describe the strength of association with 95% confidence intervals.

### Result

Among the respondents, 48.4% [95% CI: 38.6, 58.2] of health care workers intended to accept COVID-19. Intention to accept COVID-19 vaccination was significantly associated with physicians (AOR = 9.27, 95% CI: 1.27–27.32), professionals with a history of chronic illness (AOR = 4.07, 95% CI: 2.02–8.21), perceived degree of risk of COVID-19 infection (AOR = 4.63, 95% CI: 1.26–16.98), positive attitude toward COVID-19 prevention (AOR = 6.08, 95% CI: 3.39–10.91) and good preventive practices (AOR = 2.83, 95% CI: 1.58–5.08).

### Conclusion

In this study, the intention of health care workers to accept the COVID-19 vaccine was low. Professional types, history of chronic illness, perceived degree of risk to COVID-19 infection, attitude toward COVID-19 and preventive practices were found to be factors for

**Data Availability Statement:** The data underlying the results contain the potential identification of our study participants and have some ethical restrictions as set by the ethical review committee of Mizan Tepi University, College of Health

Sciences. However, the row datasets will be available from the chairman of ethics committee of college of health science, Mizan Tepi University on a reasonable request(wesenniguse770@gmail. com).

**Funding:** The authors received no specific funding for this work.

**Competing interests:** The authors have declared that no competing interests exist.

intention to accept COVID-19 vaccine in professionals. It is important to consider professional types, history of chronic illness, perceived degree of risk to COVID-19, attitude of professionals and preventive behaviors to improve the intention of professionals' vaccine acceptance.

## Introduction

Covid-19 is a pandemic acute respiratory disease that was first detected and identified in late December in China in 2019 [1]. Due to increased concerns because of cases outside China and increased incidence in China, the World Health Organization (WHO) declared the disease as a pandemic and public health emergency [2]. Since its emergence, the number of daily confirmed cases of COVID-19 worldwide has increased dramatically. The virus causes morbidity in many millions of people and has taken the lives of many millions since its emergence [3]. The situation was similar in Ethiopia; soon after the first case was detected (March 13, 2020), the incidence increased. In countries such as Ethiopia, the transmissibility of the virus is high due to overcrowding and poor socioeconomic status. The possibility of pandemics in Ethiopia is more likely due to poor infrastructure, weak health systems, large household size, inadequate sanitation, population turnover from site to site or increased mobility. The number of COVID-19 cases in Ethiopia has continued to increase. For instance, the total number of COVID-19 cases within a country by January 20, 2021, was 275,194. The mortality from the disease within a country is also high, with an estimated 3174 deaths attributed to the disease [4, 5]. However, the total number of cases detected in Ethiopia may have been underestimated because of insufficient testing capacity [6].

The pandemic is overwhelming in various economic sectors [7]. The health care sector is greatly impacted by the pandemic, with increased health care costs for medical supplies, increased demand for protective equipment, and a shortage of accommodating intensive care units and ventilation machines [8]. Front-line fighters, primarily health professionals, are at a high risk for the disease. Their susceptibility to diseases has many implications for health care systems. Their morbidity and mortality can cause severe crises in health care personnel shortages. In other words, as these professionals are always frontline for any case and are contacting clients frequently, they have the potential to infect others [7, 9].

Countries put different precautionary measures to prevent COVID-19 in accordance with WHO guidelines, including frequent hand washing, social distancing, wearing a face mask, movement limitation to crowded areas, and avoidance of consumption of raw meat to prevent cross-contamination [10]. However, adherence to these strict measures is very low in Ethiopia. A study conducted in the southern part of Ethiopia indicated that only 12.3% of the population adhered to preventive measures against COVID-19 [11].

Vaccine development was also considered to tackle the pandemic and overcome the negative consequences in different sectors [12, 13]. Despite this attempt, vaccine hesitancy and unwillingness to accept the COVID-19 vaccine is a challenge across the world [14].The unwillingness to accept the COVID-19 vaccine in Ethiopia is expected to be high. For instance, a study found that only 31.4% of the population were willing to take the COVID-19 vaccine [15]. Readiness to receive COVID-19 vaccine among professionals is also challenging; for example, only 27.7% of health care workers were intended to take COVID-19 in the Democratic Republic of the Congo. Factors identified for the intention to receive the vaccine were sex, professional type, attitude, and vaccine safety [16].

Developing countries, such as Ethiopia, are gaining COVID-19 vaccines from different donating countries to vaccinate high-risk groups such as health care professionals. Ethiopia received 2.2 million COVID-19 vaccines from the COVAX facility. Although Ethiopia is gaining vaccines, the intention of healthcare professionals to accept COVID-19 vaccination and the factors affecting it are not known. The findings from these professionals would help policy makers in the health sector to improve vaccine acceptance, which would contribute to the control of COVID-19 pandemics. Therefore, this study was undertaken to assess the intentions of health care workers and the factors associated with accepting the COVID-19 vaccine in south western Ethiopia.

## Materials and methods

### Study setting and design

The study was conducted at the Mizan Tepi University Teaching Hospital (MTUTH) and Gebre Tsadik Shawo General Hospital (GTSGH). MTUTH, located in the Bench Sheko zone, is a teaching hospital that also delivers treatment services within four adult outpatient departments (OPDs), one pediatric OPD, one emergency department, one chronic disease OPD, two adult medical wards, two adult surgical wards, one pediatric ward, one obstetrics and gynecology ward, anti-retroviral treatment, and tuberculosis treatment center. This hospital is located 568 km from the country's capital. A total of 418 health care workers were in the MTUTH. GTSGH is a public hospital located in the Kaffa zone in the southern region, located 456 km from the capital of the country. In this setting, there were 259 health care workers who delivered care in different departments. Health care workers in these settings are comprised of medical doctors, nurses, midwives, laboratory technicians, pharmacists, psychiatrists, and radiologists. A facility-based cross-sectional study was conducted among health care workers from March 15 to, 28/2021.

### Source population

All health care workers working in the Mizan Tepi University teaching and Gebre Tsadik Shawo General Hospitals were the source population.

### Study participants

Sampled health care workers from the two hospitals during the time of data collection were the study participants.

### Inclusion and exclusion criteria

Health care workers in two hospitals aged $\geq$ 18 years who were involved in the direct contacts with patients (nurses, physicians, midwifes, pharmacists, laboratory technicians, radiology technicians, psychiatry professionals) and who agreed to participate in the study were included in the study, while health care workers who were absent at the time of data collection were excluded from the study.

### Sample size and sampling technique

The sample size determined for this study was determined by a single population proportion formula, with the assumption of 50% acceptability of vaccination against COVID-19, a 95% confidence interval, 5% margin of error, and addition of 10% non-response rate. Therefore, the sample size for this study was 423. The first two hospitals, namely MTUTH and GTSGH, were selected from a total of four hospitals located in southwestern Ethiopia (Bench Sheko,

Kaffa, Sheka, and West Omo zones) using the lottery method. A predetermined sample size was allocated to each hospital. The total number of health care professionals in both hospitals was 677. Of these, 418 were in MTUTH, and 259 health care workers were in GTSGH. To obtain representative samples from both hospitals, proportional allocation was performed. Thus, 261 samples were allocated to MTUTH and 162 were allocated to GTSGH. A simple random sampling method was used to select participants from both hospitals.

## Data collection tool, quality control, and procedure

The tool used to collect data for this study was developed and designed in a local context after reviewing relevant studies [16–20]. The tool was designated in English and provided to the participants, as all participants could understand the questionnaire well since the working and educational language was English. The tool essentially contained seven parts; part I assessed socio-demographic characteristics, part II assessed health status and COVID-19 experience, part III assessed participants' knowledge about COVID-19, part IV assessed participants' attitude towards COVID 19, part V assessed COVID-19 Prevention Practices among healthcare workers, part VI assessed vaccine hesitancy and part VII assessed willingness to accept COVID-19 vaccine. A one-day orientation of data collection was given to four lecturers who collected the data. Pretesting was performed on 21 health care workers working in Wacha Hospital, which is different from the actual study sites. Based on the pretest results, necessary modifications were made to the questionnaires. The questionnaire was self-administered (paper survey) and all data collectors and supervisors strictly adhered to the WHO and national standards of COVID-19 prevention protocols. The data collectors were worn face masks, gloved their hands, maintained distance, and sanitized their hands between each questionnaire administration.

Nineteen items were used to assess the knowledge of health care workers regarding COVID-19. The questions focused on the clinical manifestation, case suspicion, incubation period, mode of transmission, and prevention methods. Participants' responses related to knowledge items were scored by assigning one for correct answers and zero for incorrect answers including "I don't know" responses. In such a way that participants' knowledge score may range from zero to nineteen. To categorize participants as having good knowledge and poor knowledge, the knowledge means sore was computed and participants with knowledge scores greater than or equal to the knowledge mean score were considered as having good knowledge and otherwise having poor knowledge [21].

Seven items were used to assess the attitudes of the participants toward COVID-19 preventive measures. Participants' responses were scored by giving one point to "yes" and zero to "no" and "not sure" responses. Responses to reversed question was reversed when assigning the points (yes = zero, not sure = zero and no = one). In this way, participants' total attitude scores ranged from zero to seven [22]. The mean attitude score was computed and participants were considered to have a positive attitude if the attitude score was ≥ mean attitude score [21]. The internal consistency of seven attitude items was checked and Cronbach's alpha became 0.818.

The practice was assessed by five questions and scored by assigning one point to a yes response and zero to no response. The mean practice score was computed and participants were considered to have good practice if the practice score was greater than or equal to the mean practice score [21].

We evaluated self-reported vaccine hesitancy of participants according to the WHO definition by three adapted questions asking, "Have you ever refused a vaccine for yourself or a child because you considered it as useless or dangerous?" "Have you ever postponed a vaccine

recommended by a physician because of doubts about it?", and "Have you ever had a vaccine for a child or yourself despite doubts about its efficacy [23]?" If participants' responses were yes to at least one of the three questions, participants were considered vaccine-hesitant.

HCWs' intention to accept the COVID-19 vaccine was assessed by one question asking "will you get the COVID-19 vaccine if it is available?" Participants' responses were dichotomized into "yes" and "no."

### Data processing and analysis

Data were entered into Epidata version 3.1, after a manual check for completeness. The entered data were exported to SPSS version 23 and both descriptive and inferential statistics were used. Statistical significance was set at p < 0.25 in bivariate logistic regression analysis to identify candidate variables for multivariable logistic regression analysis. In the multivariable analysis, a significant association was found with a p-value of less than 0.05. The associations were presented with an adjusted odds ratio (AOR) and corresponding 95% CI.

### Ethical considerations

This study was approved by the ethical review committee of the Mizan Tepi University, College of health sciences. The letter was submitted to both hospital administrates to begin the study. The confidentiality of the respondents was secured by excluding respondent's identifiers such as names from the data collection format. Written Informed consent was obtained from all the participants. The right of participants to withdraw from the study at any time was clearly stated for the participants.

## Results

### Socio-demographic characteristics of the respondents

A total of 405 filled self-administered questionnaires were returned with a response rate of 96.0%. The majority of the respondents were below the age of 30 (72.5%), male (50.4%), and married (57.3%). Of the total HCWs, 59.8% were Nurses and 65.2% were first-degree holders. Regarding economic status, 55.6% of the participants earned 91.3–182.4 USD (Table 1).

### Health status and COVID-19 experience of health care workers

Among the respondents, 285(70.4%) had no history of chronic illness. The majority, 337 (83.2%) perceived that they were at a higher risk of COVID-19 infection, and 9(2.2%) reported a previous COVID-19 infection. In addition, 209(51.6%) knew of friends, neighbors, or colleagues infected by coronaviruses (Table 2).

### COVID-19 knowledge of the health care workers

The mean knowledge score was 13.62 (±3.77). More than half, 249(61.5%), of the respondents, had good knowledge about COVID-19 whereas, 156(38.5%) had poor knowledge. More than two-thirds of the study participants, 321 (79.3%), the SARS-COV-2 virus spreads via the respiratory droplets of an infected individual. The majority, 331(81.7%), knew that the main symptoms of COVID-19 were fever, fatigue, dry cough, and myalgia. In addition, 253(62.5%) of them mentioned that PPE such as respiratory protection, cannot effectively protect users from COVID-19 unless it is properly and consistently worn (Table 3).

**Table 1. Socio-demographic characteristics of the health care workers, hospitals of south western Ethiopia, 2021 (n = 405).**

| Variables | Category | Frequency | Percentage |
|---|---|---|---|
| Age category (in years) | <30 | 294 | 72.6 |
| | 31–40 | 95 | 23.5 |
| | 41–50 | 8 | 2.0 |
| | >60 | 8 | 2.0 |
| Sex | Male | 204 | 50.4 |
| | female | 201 | 49.6 |
| Marital status | Single | 156 | 38.5 |
| | married | 232 | 57.3 |
| | other[1] | 17 | 4.2 |
| Profession type | Physicians | 25 | 6.2 |
| | Nurse | 242 | 59.8 |
| | Midwifery | 52 | 12.8 |
| | Medical laboratory | 34 | 8.4 |
| | Pharmacist | 34 | 8.4 |
| | Others[2] | 18 | 4.4 |
| Highest qualification level | Diploma | 127 | 31.4 |
| | Degree | 264 | 65.2 |
| | Masters | 14 | 3.5 |
| Monthly salary in USD | 68.4–91.2 | 149 | 36.8 |
| | 91.3–182.4 | 225 | 55.6 |
| | >182.4 | 31 | 7.7 |
| Number of people in a household | 1 | 138 | 34.1 |
| | 2 | 71 | 17.5 |
| | 3–4 | 157 | 38.8 |
| | 5–6 | 29 | 7.2 |
| | >7 | 10 | 2.5 |

[1] Widowed and Divorced

[2] Radiology technicians, Psychiatry professionals

**Table 2. Health status and COVID-19 experience of health the professionals working at hospitals of southwestern Ethiopia, 2021 (n = 405).**

| Variables | Category | Frequency | Percentage |
|---|---|---|---|
| Previously diagnosed with chronic diseases | Yes | 120 | 29.6 |
| | No | 285 | 70.4 |
| Do you have any of the following diseases? (Type 2 diabetes mellitus, Chronic Obstructive Pulmonary Disease (COPD), Cancer, Kidney Failure, Heart diseases, Sickle Cell Anemia) | Yes | 78 | 19.3 |
| | No | 327 | 80.7 |
| Do you have any of the following diseases? (Type 1 diabetes mellitus, Hypertension, Bone marrow transplant, Cerebrovascular diseases or stroke, Cystic Fibrosis, Asthma, Taking steroids or immunosuppressant drugs, Hepatic diseases, Thalassemia, Lung fibrosis) | Yes | 52 | 12.8 |
| | No | 353 | 87.2 |
| Perceived risk to COVID-19 infection | High | 337 | 83.2 |
| | Medium | 34 | 8.4 |
| | Low | 34 | 8.4 |
| Personal history of COVID-19 infection | Yes | 9 | 2.2 |
| | No | 396 | 97.8 |
| Know any friends, neighbors, or colleagues infected by Coronavirus | Yes | 196 | 48.4 |
| | No | 209 | 51.6 |

**Table 3. COVID-19 knowledge of the HCWs working at hospitals of southwestern Ethiopia, 2021 (n = 405).**

| Variables | Category | Frequency | Percentage |
|---|---|---|---|
| A suspected case is a patient with acute respiratory illness and recent history of travel. | True | 289 | 71.4 |
| | False | 61 | 15.1 |
| | I don't know | 55 | 13.6 |
| A person with laboratory confirmation of COVID 19 infection, irrespective of clinical signs and symptoms is a confirmed case. | True | 305 | 75.3 |
| | False | 65 | 16 |
| | I don't know | 35 | 8.6 |
| A suspected case is any patient with fever and at least cough or shortness of breath. | True | 372 | 91.9 |
| | False | 29 | 7.2 |
| | I don't know | 4 | 1 |
| Any patient with a history of contact with a confirmed or probable COVID 19 case in the last 14 days before symptom onset is a suspected case. | True | 312 | 77 |
| | False | 40 | 9.9 |
| | I don't know | 53 | 13.1 |
| The main clinical symptoms of COVID-19 are fever, fatigue, dry cough, and myalgia | True | 331 | 81.7 |
| | False | 68 | 16.8 |
| | I don't know | 6 | 1.5 |
| Unlike symptoms of common cold, stuffy nose, running nose, and sneezing are less common in persons infected with the SARS-COV-2. | True | 216 | 53.3 |
| | False | 95 | 23.5 |
| | I don't know | 94 | 23.2 |
| Eating Monkey, Bat or contacting wild animals would result in the infection by the SARS-COV-2. | True | 262 | 64.7 |
| | False | 106 | 26.2 |
| | I don't know | 37 | 9.1 |
| Patients with COVID-19 cannot spread the virus to others when they do not show signs and symptoms of the disease. | True | 131 | 32.3 |
| | False | 252 | 62.2 |
| | I don't know | 22 | 5.4 |
| The SARS-COV-2 virus spreads via respiratory droplets of infected individual. | True | 321 | 79.3 |
| | False | 42 | 10.4 |
| | I don't know | 42 | 10.4 |
| The incubation period of COVID-19 lasts up to 14days. | True | 358 | 88.4 |
| | False | 30 | 7.4 |
| | I don't know | 17 | 4.2 |
| Children and young adults are less likely to be infected with COVID 19 thus, precautionary measures are not necessary to prevent the infection. | True | 197 | 48.6 |
| | False | 190 | 46.9 |
| | I don't know | 18 | 4.4 |
| Not all patients infected with COVID-19 will develop severe cases. | True | 265 | 65.4 |
| | False | 128 | 31.6 |
| | I don't know | 12 | 3 |
| Patients with underlying chronic disease conditions are at higher risk of infection and death from COVID 19. | True | 350 | 86.4 |
| | False | 36 | 8.9 |
| | I don't know | 19 | 4.7 |
| Avoiding handshakes, crowded places, and public transportation could help to prevent COVID-19 | True | 317 | 78.3 |
| | False | 56 | 13.8 |
| | I don't know | 32 | 7.9 |
| Antibiotics are the first line of treatment when you suspect or have a confirmed case of COVID-19. | True | 139 | 34.3 |
| | False | 244 | 60.2 |
| | I don't know | 22 | 5.4 |

(*Continued*)

**Table 3.** (Continued)

| Variables | Category | Frequency | Percentage |
|---|---|---|---|
| Early recognition and supportive treatment help most patients recover from the infection since there is no effective cure for COVID-19. | True | 282 | 69.6 |
| | False | 97 | 24 |
| | I don't know | 26 | 6.4 |
| Isolation and treatment of people who are infected with the COVID-19 virus are effective ways to break the chain of transmission. | True | 316 | 78 |
| | False | 50 | 12.3 |
| | I don't know | 39 | 9.6 |
| Personal protective equipment (PPE) like respiratory protection cannot effectively protect the users if it is not properly and consistently worn. | True | 253 | 62.5 |
| | False | 100 | 24.7 |
| | I don't know | 52 | 12.8 |
| Wearing face masks are used to protect both the Health care Workers and the patient. | True | 282 | 69.6 |
| | False | 69 | 17 |
| | I don't know | 54 | 13.3 |
| Overall knowledge status | Good knowledge | 249 | 61.5 |
| | Poor knowledge | 156 | 38.5 |

## Attitude towards COVID-19 preventive measures

As stated in the methodology, the mean positive response was 4.9 (±1.6). Of the respondents, 273(65.6%) had a positive attitude toward COVID-19 prevention whereas, 143(34.4%) had a negative attitude. The majority, 364(89.9%), believed that social distancing and hand washing could prevent COVID-19 while 171(41.1%) reported that they had attended social events recently. Of the study participants, 233(57.5%) believed that the COVID-9 vaccine could prevent infection. Most of the HCWs, 321(79.3%) were confident in providing care to a suspected case of COVID-19 and 252(62.2%) said that the current preventive measures put in place by the Government could mitigate COVID-19 (Table 4).

## COVID-19 prevention practices among health care workers

The mean practice score was 3.9 (±1.3). Two hundred and seventy-nine (67.1%) respondents had good COVID-19 prevention practices. The majority, 356(87.9%) washed or sanitized their hands regularly and 327(80.7%) wore facemasks regularly at the point of care for the sick patients (Table 5).

## Vaccine hesitancy and intention to accept COVID-9 vaccine

Among the HCWs, 212(52.3%) were found to be vaccine-hesitant (Table 6). The study showed that 196 (48.4% (95% CI: 38.6, 58.2)) of health care workers would intend to accept the COVID-19 vaccine, and the remaining 209 (51.6%) intended to not accept the vaccine.

## Factors associated with HCW's intention to accept COVID-19 vaccine

Only variables with p < 0.25 during bivariate analyses were entered in multivariable analysis. In the multivariable analysis type of profession, previous personal history of chronic illness, perceived degree of risk to COVID-19, Attitude toward COVID-19, and preventive practice toward COVID-19 were found to be significantly associated with HCW's intention to accept the COVID-9 vaccine. According to this study's findings, health care workers with a physician profession were 9 times more likely to have the intention to accept the COVID-19 vaccine

**Table 4. Attitude towards COVID-19 prevention among HCWs working at hospitals of southwestern Ethiopia, 2021 (n = 405).**

| Variables | Category | Frequency | Percentage |
|---|---|---|---|
| Do you believe that social distancing and hand washing could prevent COVID-19? | Yes | 364 | 89.9 |
| | No | 35 | 8.6 |
| | Not sure | 6 | 1.5 |
| Do you have confidence in the current preventive measures put in place by the Government to mitigate COVID 19? | Yes | 252 | 62.2 |
| | No | 88 | 21.7 |
| | Not sure | 65 | 16.0 |
| Have you attended any social events recently?* | Yes | 171 | 41.1 |
| | No | 205 | 49.3 |
| | Not sure | 29 | 7.0 |
| Are you confident to provide care to a suspected case of COVID 19? | Yes | 321 | 79.3 |
| | No | 63 | 15.6 |
| | Not sure | 21 | 5.2 |
| Do you believe that the COVID-9 vaccine can prevent infection? | Yes | 233 | 57.5 |
| | No | 110 | 27.2 |
| | Not sure | 62 | 15.3 |
| Health insurance or incentives can motivate health care workers directly involved in the management of COVID 19 patients? | Yes | 300 | 74.1 |
| | No | 83 | 20.5 |
| | Not sure | 22 | 5.4 |
| Are you ready to participate in community sensitization on COVID 19? | Yes | 311 | 76.8 |
| | No | 53 | 13.1 |
| | Not sure | 41 | 10.1 |
| Overall attitude status | Positive attitude | 273 | 65.5 |
| | Negative attitude | 143 | 34.4 |

*Reversed item and scoring was reversed for this item.

than nurses (AOR = 9.27, 95% CI: 1.27–27.32). Health care workers with a history of chronic illness were 4 times more likely to have the intention to accept the COVID-19 vaccine than health care workers without a history of chronic illness (AOR = 4.07, 95% CI: 2.02–8.21) and health care workers who perceived their degree of risk medium were 5 times more likely to

**Table 5. COVID-19 prevention practices among HCWs working at hospitals of southwestern Ethiopia, 2021 (n = 405).**

| Variables | Category | Frequency | Percentage |
|---|---|---|---|
| Do you wash your hands or sanitize your hands regularly? | Yes | 356 | 87.9 |
| | No | 49 | 12.1 |
| Do you regularly use facemask at point of care (when rendering service to sick patients) | Yes | 327 | 80.7 |
| | No | 78 | 19.3 |
| Do you use a facemask when you have flu-like symptoms? | Yes | 316 | 78.0 |
| | No | 89 | 22.0 |
| Do you use non-conventional remedies (Honey, garlic, ginger, and lime) when you have flu-like symptoms? | Yes | 263 | 64.9 |
| | No | 142 | 35.1 |
| In recent times, have you worn a face mask when leaving your home? | Yes | 310 | 76.5 |
| | No | 95 | 23.5 |
| Overall Practice status | Good Practice | 279 | 67.1 |
| | Poor Practice | 137 | 32.9 |

**Table 6. Vaccine hesitancy among HCWs working at hospitals of southwestern Ethiopia, 2021 (n = 405).**

| Variables | Response | Frequency | Percentage |
|---|---|---|---|
| Have you ever refused a vaccine for yourself or a child because you considered it useless or dangerous? | Yes | 67(16.5) | 16.5 |
| | No | 338(83.5) | 83.5 |
| Have you ever postponed a vaccine recommended by a physician? | Yes | 91(22.5) | 22.5 |
| | No | 314(77.5) | 77.5 |
| Have you ever had a vaccine for a child or yourself despite doubts about its efficacy? | Yes | 91(22.5) | 22.5 |
| | No | 314(77.5) | 77.5 |
| Vaccine hesitant | Yes | 212(52.3) | 52.3 |
| | No | 193(47.7) | 47.7 |

have intention to accept COVID-19 vaccine than health care workers who perceived their degree of risk to be low (AOR = 4.63, 95% CI: 1.26–16.98). Intention to accept the vaccine was 3 times more likely among those with good preventive practices than among those with poor practice (AOR = 2.83, 95% CI: 1.58–5.08), and the intention to accept was 6 times more likely among those with a positive attitude toward COVID-19 than among those with negative attitudes (AOR = 6.08, 95% CI: 3.39–10.91) (Table 7).

## Discussion

Health care workers (HCWs) are the frontlines in combating COVID-19 infection, which makes them more vulnerable to infection than other parts of the society [24, 25]. Since the discovery of the novel coronavirus infection, thousands of health professionals have been infected and lost their lives because of the disease worldwide [26]. In this study, most respondents (83.2%) also perceived that they were at a higher risk of COVID-19 infection. Thus, it is crucial to implement preventive measures including vaccinations against the virus. Therefore the current study focused on the intention of HCWs to accept or not accept the COVID-19 vaccine.

According to the current study, most HCWs (61.5%) possessed good knowledge about COVID-19. This finding was lower than that of two studies in Ethiopia [27, 28], and that of Pakistan [29] in which previous studies stated that more than three quarters of the health professionals had satisfactory knowledge. This discrepancy might be due to differences in the methodology and study settings. However, the knowledge level of the respondents did not show an association with acceptance of the COVI-19 vaccine.

In this study 52.3% of HCWs were vaccine-hesitant and among respondents, 48.4% would intend to accept a COVID-19 vaccine. However, no association was found between vaccine hesitancy and the intention to accept the COVID 19-vaccine in the present study. This is not supported by a previous study from France which, concluded that there was a significant association between vaccine hesitancy and acceptance [30]. The possible reason for the vaccine hesitancy among health care workers in the present study might be vaccine misinformation about the adverse effects of the COVID-19 vaccine. This perception is one of the obstacles in accepting the vaccine [31]. The proportion of health care workers who intended to accept the COVID-19 vaccine was different from the study finding from France [30], which indicated that the COVID vaccine acceptance rate among health care providers was 76.9%. A possible reason for the discrepancy between the current and the previous study might be due to the difference in the study setting and the previous study was conducted a few months after the discovery of the disease but before the COVID-vaccine was introduced to the world.

The study also pointed out that the intention of nurses to accept the COVID-19 vaccine (44.6%) was relatively lower than other professionals such as pharmacists (52.9%), physicians

**Table 7. Factors associated with health care workers intention to accept COVID-19 vaccine in southwestern Ethiopia, 2021.**

| Variables | Intention to accept COVID-19 vaccine | | COR (95%CI) | AOR (95%CI) |
|---|---|---|---|---|
| | **No** | **Yes** | | |
| **Type of profession** | | | | |
| Physician | 4(16%) | 21(84%) | 6.51(2.17–11.55) | 9.27(1.27–27.32)* |
| Midwifery | 21(40.4%) | 31(59.6%) | 1.83(0.29–3.37) | 2.44(0.38–5.34) |
| Medical laboratory | 22(64.7%) | 12(35.3%) | 0.68 (0.32–1.43) | 0.67(0.28–1.59) |
| Pharmacist | 16(47.1%) | 18(52.9%) | 1.40(0.68–2.87) | 1.28(0.47–3.49) |
| Others | 12(66.7%) | 6(33.7%) | 0.62(0.22–1.71) | 1.23(0.35–4.33) |
| Nurse | 134(55.4%) | 108(44.6%) | 1 | 1 |
| **Monthly income** | | | | |
| 68.4–91.2 USD | 77(51.7%) | 72(48.3%) | 0.44(0.19–1.01) | 1.65(0.31–8.83) |
| 91.3–182.4 USD | 122(54.2%) | 103(45.8%) | 0.40(0.18–0.89) | 1.04(0.20–5.42) |
| > 182.4 USD | 10(32.3%) | 21(67.7%) | 1 | 1 |
| **Previously diagnosed with chronic illness** | | | | |
| Yes | 39(32.5%) | 81(67.5%) | 3.07(1.96–4.81) | 4.07(2.02–8.21)* |
| No | 170(59.6%) | 115(40.4%) | 1 | 1 |
| **Perceived degree of risk to COVID-19** | | | | |
| High | 182(54.0%) | 155(46.0%) | 0.85(0.42–1.72) | 2.39(0.88–6.51) |
| Medium | 10(29.4%) | 24(70.6%) | 2.40(0.88–6.51) | 4.63(1.26–16.98)* |
| Low | 17(50.0%) | 17(50.0%) | 1 | 1 |
| **Know any friends, neighbors, or colleagues infected by COVID-19** | | | | |
| Yes | 90(45.9%) | 106(54.9%) | 1.56(0.05–2.31) | 1.70(0.27–2.87) |
| No | 119(56.9%) | 90(43.1%) | 1 | 1 |
| **Attitude towards COVID-19** | | | | |
| Positive Attitude | 108(76.5%) | 165(23.5%) | 4.98(3.11–7.96) | 6.08(3.39–10.91)* |
| Negative Attitude | 101(76.5%) | 31(23.5%) | 1 | 1 |
| **COVID-19 prevention Practice** | | | | |
| Good practice | 128(45.9%) | 151(54.1%) | 2.12(1.38–3.28) | 2.83(1.58–5.08)* |
| Poor practice | 81(64.3%) | 45(35.7%) | 1 | 1 |
| **Vaccine hesitancy** | | | | |
| Yes | 116(54.7%) | 96(45.3%) | 0.77(0.52–1.14) | 0.77(0.48–1.22) |
| No | 93(48.2%) | 100(51.8%) | 1 | 1 |

*Significant at p-value < 0.05

(84%), and midwifery (59.6%). This finding is supported by a previous study that showed a lower rate of COVID-19 vaccine acceptance among nurses [30]. The lower acceptance rate in this profession is concerning because nurses are the largest workforce in the healthcare setup, have frequent contact with patients and spend more time caring than other professional categories [32].

In the current study, it was found that the intention to accept the COVID-19 vaccine was nearly fifteen times more likely among physicians than among other health professionals (radiology technicians and psychiatry). This finding was in line with another similar study finding that physicians were more prone to accept vaccination against COVID-19 than other health professionals [30]. HCWs are among the most affected groups of the population which makes them more sensitive to preventive measures particularly, the COVID-19 vaccine [24, 25]. Moreover, HCWs like physicians possess deeper knowledge about the disease and its vaccine

than other parts of the population [33]. Physicians may have observed the fatality of the disease which may increase the odds of physicians having the intention to accept a COVID-19 vaccine.

COVID-19 vaccine acceptance was more likely among respondents who said their degree of risk to COVID-19 infection as 'medium' than who said 'low'. This result was supported by another study finding that mentioned fear about COVID-19 and self-perceived risk of coronavirus infection were associated with COVID-19 vaccine acceptance among health workers [30]. In addition, healthcare workers with a history of chronic illness had increased odds of having the intention to accept the COVID-19 vaccine than participants without chronic illness. A possible reason might be due to the reason that, most (86.4%) of the study participants knew that patients with underlying chronic disease conditions were at a higher risk of infection and death from COVID-19.

In the current study, two-thirds of the respondents (65.6%) had a positive attitude toward COVID-19 prevention, and 57.5% believed that the COVID-9 vaccine could prevent infection. Intention to accept the COVID-19 vaccine was more likely among those with a positive attitude toward COVID-19 prevention than their counterparts. In line with this finding, studies have indicated that COVID-19 vaccine hesitancy is associated with a negative attitude toward COVID-19 and its preventive measures [34].

Among the study participants, 67.1% had good COVID-19 prevention practices, 87.9% washed or sanitized their hands regularly and 80.7% wore facemasks regularly at the point of care for sick patients. Intention to accept the vaccine was found to be more likely among those with good preventive practices.

## Conclusion

The intention to accept the COVID-19 vaccine was relatively low. In addition, the vaccine hesitancy observed in this study was high. HCWs' intention to accept the COVID-19 vaccine was significantly associated with the type of health professionals, personal history of chronic illness, perceived degree of risk to COVID-19 infection, attitude toward COVID-19 prevention, and preventive practice. Strategies for enhancing the acceptance of the COVID-19 vaccine by considering categories of professionals, chronic illness history, perceived risks of disease, attitude towards disease prevention and preventive behaviors among health workers are crucial. Health sector managers should stress awareness creation to alleviate misinformation about COVID-19 vaccines in health professionals to overcome COVID-19 hesitancy through different strategies.

## Limitation of the study

Our study has the following limitations; first, the study was cross-sectional and couldn't identify causality. Second, the study was conducted in governmental hospitals and may not represent health care workers outside the governmental hospital (private hospitals). Third, the study is conducted in two hospitals out of hospitals found in southwestern Ethiopia and the study explores the attitudes in these two hospitals. Despite these limitations the study highlights the intention of health care workers and associated factors to accept the COVID-19 vaccine in Ethiopian health care workers.

## Supporting information

**S1 Questionnaire. English version of the survey questionnaire.**
(DOCX)

## Acknowledgments

We would like to thank all the respondents, data collectors, and supervisors for the realization of this study.

## Author Contributions

**Conceptualization:** Abiy Tadesse Angelo, Daniel Shiferaw Alemayehu, Aklilu Mamo Dachew.

**Data curation:** Abiy Tadesse Angelo, Daniel Shiferaw Alemayehu, Aklilu Mamo Dachew.

**Formal analysis:** Abiy Tadesse Angelo, Daniel Shiferaw Alemayehu, Aklilu Mamo Dachew.

**Investigation:** Abiy Tadesse Angelo, Daniel Shiferaw Alemayehu, Aklilu Mamo Dachew.

**Methodology:** Abiy Tadesse Angelo, Daniel Shiferaw Alemayehu, Aklilu Mamo Dachew.

**Resources:** Abiy Tadesse Angelo, Daniel Shiferaw Alemayehu, Aklilu Mamo Dachew.

**Software:** Abiy Tadesse Angelo, Daniel Shiferaw Alemayehu, Aklilu Mamo Dachew.

**Supervision:** Abiy Tadesse Angelo, Daniel Shiferaw Alemayehu, Aklilu Mamo Dachew.

**Validation:** Abiy Tadesse Angelo, Daniel Shiferaw Alemayehu, Aklilu Mamo Dachew.

**Visualization:** Abiy Tadesse Angelo, Daniel Shiferaw Alemayehu, Aklilu Mamo Dachew.

**Writing – original draft:** Abiy Tadesse Angelo, Daniel Shiferaw Alemayehu, Aklilu Mamo Dachew.

**Writing – review & editing:** Abiy Tadesse Angelo, Daniel Shiferaw Alemayehu, Aklilu Mamo Dachew.

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
