## [Decision Letter · Decision Letter 0]

22 Jun 2021

PONE-D-21-13549

Health Care Workers Intention to Accept COVID-19 Vaccine and Associated Factors in Southern Western Ethiopia, 2021

PLOS ONE

Dear Dr. Angelo,

Thank you for submitting your manuscript to PLOS ONE. After careful consideration, we feel that it has merit but does not fully meet PLOS ONE’s publication criteria as it currently stands. Therefore, we invite you to submit a revised version of the manuscript that addresses the points raised during the review process.

We look forward to receiving your revised manuscript.

Kind regards,

Livia Melo Villar

Academic Editor

PLOS ONE

Additional Editor Comments:

Dear Author,

I have read the paper and comments of the reviewers. Based on these comments, I suggest major revision of the paper,

Sincerely,

Livia

Journal Requirements:

3. Please include additional information regarding the tool or questionnaire used in the study and ensure that you have provided sufficient details that others could replicate the analyses. For instance, if you developed the tool or questionnaire as part of this study and it is not under a copyright more restrictive than CC-BY, please include a copy, in both the original language and English, as Supporting Information. If the questionnaire is published, please provide a citation to the (1) questionnaire and/or (2) original publication associated with the questionnaire.

Reviewers' comments:

Reviewer's Responses to Questions

**Comments to the Author**

1. Is the manuscript technically sound, and do the data support the conclusions?

Reviewer #1: Yes

Reviewer #2: Yes

Reviewer #3: Partly

2. Has the statistical analysis been performed appropriately and rigorously? 

Reviewer #1: Yes

Reviewer #2: Yes

Reviewer #3: No

3. Have the authors made all data underlying the findings in their manuscript fully available?

Reviewer #1: Yes

Reviewer #2: Yes

Reviewer #3: Yes

4. Is the manuscript presented in an intelligible fashion and written in standard English?

Reviewer #1: Yes

Reviewer #2: Yes

Reviewer #3: No

5. Review Comments to the Author

Reviewer #1: Dear Editor,

The manuscript entitled “Health Care Workers Intention to Accept COVID-19 Vaccine and Associated Factors in Southern Western Ethiopia, 2021” by Angelo et al. has merit, However, some points should be addressed to improve the quality of the paper.

Abstract:

The conclusion repeated some the results but did not provide new insights or ideas on the topic addressed.

Introduction:

Page 10, line 3: Please, correct the sentence “...cases outside a China”

Page 11, line 8-10 (last paragraph of ‘introduction’ section): Please, make clear the aouthor’s hipothesis.

Material and Methods:

What abouth the ethics? Please, provide more informations.Did the participants sign the free informed consent?

Page 12, ‘Sample Size and Sampling Technique’ section: How do the authors estimated the vaccine acceptance in 50%? Please, provide the reference for this assumption.

Page 13, lines 2-3: The sentence " The lists of all health care workers were obtained from both hospitals, and the lottery method was used to select 261 samples from MTUTH and 162 from GTSGH.” is redundant. Please, remove it.

Results:

‘Sociodemographic characteristics of the respondents” section and table 1: Please, present the participant’s monthly salary in US dollars. It is more informative.

Discussion:

It would be interesting to deepen the discussion about if (and how) socio-cultural characteristics of the population could be related to the hesitancy to take the COVID-19 vaccine. In addition, insights on how to overcome this hesitancy should be proposed at the end of the discussion or in the conclusion.

In addition, some formatting errors, such as parentheses and capital letters, need to be corrected

Reviewer #2: Intro:

It would be nice to have more data regarding the current situation in Ethiopia and the situation around the time of the study. Had cases continued to rise at that time? How bad was the area surveyed hit?

Are there any data regarding public immunization willingness in Ethiopia?

How well did the residents of Ethiopia accept public health recommendations for things like masks etc? This would help better frame the research.

Also- what is the current vaccine avaialbility in Ethiopia? Are any vaccine available?

General: Recommend better editing for English throughout.

COVID-19 should be capitalized throughout

Abstract: Methods should be clear that this is a survey of healthcare workers at two hospitals.

Methods: How was the survey conducted? Was it anonymous? Was it a paper survey or administered in person?

What types of healthcare workers were eligible? Beyond Nurses, what other types of healthcare workers were eligible to be included? Please also give more information on what "first degree holders" are. Given that "Others" are the reference population, we need more information about that others" actually are. Consider making nurses or a different, larger, group the reference group.

Reviewer #3: This cross-sectional study examined the prevalence and factors associated with acceptance of COVID-19 vaccine in 405 health care workers recruited in 2 public hospitals in Southern Ethiopia in March 2021. I have the following comments for the authors to consider and improve the analyses and the clarity of the findings:

1. The authors are suggested to give more details on the context/ development of COVID-19 in (Southern) Ethiopia near the data collection period (March 2021), which could influence HCW’s acceptance of the vaccine.

2. The scoring of COVID-19 attitude seemed problematic. I could not understand why the responses were ranked from positive (3 points), negative (2 points) to not sure (1 point). Should “not sure” score 2 points while negative 1 point? Furthermore, since there were 7 items, the total scores should range from 7 to 21, not 7 to 18. Can the authors show the internal consistency of these 7 items? In the Methods, please also make it clear that these measures are referring to attitude toward COVID-19 preventive measures.

3. Responses to the question of COVID-19 vaccine hesitancy were dichotomised to “yes” and “no”. Can the authors presented the original scale of the responses?

4. The logistic regression models used in the study appeared to be “multivariable” (1 dependent variable, multiple independent variables) instead of “multivariate” (multiple dependent variables). Please correct.

5. In the abstract the sample size was 423, whereas in the main text it was 405. Please correct.

6. Table 2: Was there any particular reason to divide the chronic diseases into the two categories listed?

7. Table 4: Please indicates the correct answers for the questions on COVID-19 knowledge

8. Table 5: I have doubts on the question “Do you use non-conventional remedies (Honey, garlic, ginger, and lime) when you have flu-like symptoms?” as a measure of COVID-19 prevention practice. This dose not seem a proven/ recommended practice to prevent COVID-19.

9. Please shows the overall prevalence with 95% CI of COVID-19 vaccine acceptance in the results and in the abstract.

10. Table 7: The percentages shown appeared to be cell percentages (rows and columns add up to 100%), which are not useful nor straightforward to interpret. The authors should use row percentages (percentages in the same row add up to 100%) to show the prevalence of vaccine acceptance in each subgroup of variables. Also, it seems more meaningful to use “Nurse” as the reference group instead of “Others” for the variable “Type of Profession”. Please also show the association of vaccine hesitancy and COVID-19 vaccine acceptance (not presented in the table) and which variables were adjusted in the adjusted models.

11. Discussion 3rd paragraph: It is surprising to see that vaccine hesitancy was not associated with COVID-19 vaccine acceptance, since many prior studies have shown vaccine hesitancy/ previous vaccination history to be strongly associated with vaccine acceptance. I could not interpret the results because they were not presented (see comment 10).

12. The vaccine acceptance among nurses appeared to rather low relative to other healthcare workers (~45% vs 54% based on the numbers presented in table 7). This is concerning because nurses constitute the largest health care workforce and have frequent contacts with patients. Can the authors discuss the finding?

13. The authors should elaborate the implications of their findings/ How their findings could be used to improve COVID-19 vaccine acceptance?

14. The authors should discuss the limitations of their findings, which are absent in the article

15. The article would be much benefited from English editing.

6. PLOS authors have the option to publish the peer review history of their article (what does this mean?). If published, this will include your full peer review and any attached files.

Reviewer #1: No

Reviewer #2: No

Reviewer #3: **Yes: **TT Luk

---

## [Author Response · Author response to Decision Letter 0]

7 Jul 2021

We would like to extend our thanks for valuable comments from the editor.

 We have ensured that our manuscript is written in the journal’s style requirements, including for file naming.

2. Please provide additional details regarding participant consent. In the ethics statement in the Methods and online submission information, please ensure that you have specified what type you obtained (for instance, written or verbal, and if verbal, how it was documented and witnessed). If your study included minors, state whether you obtained consent from parents or guardians. If the need for consent was waived by the ethics committee, please include this information. Once you have amended this/these statement(s) in the Methods section of the manuscript, please add the same text to the “Ethics Statement” field of the submission form (via “Edit Submission”).

 We have added additional details regarding participant consent in revised manuscript. Our study doesn’t include minors and we have added “ethics statement” during the submission of the revised manuscript.

3. Please include additional information regarding the tool or questionnaire used in the study and ensure that you have provided sufficient details that others could replicate the analyses. For instance, if you developed the tool or questionnaire as part of this study and it is not under a copyright more restrictive than CC-BY, please include a copy, in both the original language and English, as Supporting Information. If the questionnaire is published, please provide a citation to the (1) questionnaire and/or (2) original publication associated with the questionnaire.

 We have uploaded questionnaire as supporting information.

 The data underlying the results contain the potential identification of our study participants and have some ethical restrictions as set by the ethical review committee of Mizan Tepi University, College of Health Sciences. However, the row datasets will be available on reasonable request after requesting abiyutad@mtu.edu.et .

Comments to reviewers

Reviewer #1

1. The conclusion repeated some the results but did not provide new insights or ideas on the topic addressed.

 Thank you reviewer for this important observation. We have made conclusion based on your comments in the revised manuscript.

2. Introduction Page 10, line 3: Please, correct the sentence “...cases outside a China” 

 We have corrected the sentences with in the revised manuscript.

3. Page 11, line 8-10 (last paragraph of ‘introduction’ section): Please, make clear the authors' hypothesis.

 Good observation dear reviewer. We have corrected it accordingly.

4. What about the ethics? Please, provide more information. Did the participants sign the free informed consent?

 Thank you reviewer for such comments, and we have provided more information like the form of informed consent in the revised manuscript.

5. Page 12, ‘Sample Size and Sampling Technique’ section: How do the authors estimated the vaccine acceptance in 50%? Please, provide the reference for this assumption

 We used the proportion of vaccine acceptance in professionals as 50% because; during sample size estimation we didn’t find any study that identified the proportion of COVID-19 vaccine acceptance in Ethiopia. Therefore we used a proportion of vaccine acceptance of 50 % which could provide the minimum sample size for our study. 

6. Page 13, lines 2-3: The sentence “The lists of all health care workers were obtained from both hospitals, and the lottery method was used to select 261 samples from MTUTH and 162 from GTSGH.” is redundant. Please, remove it.

 Thank you for good observation, and we have removed the redundant sentence.

7. ‘Socio-demographic characteristics of the respondents” section and table 1: Please, present the participant’s monthly salary in US dollars. It is more informative.

 We have changed the monthly salary of professionals stated in Ethiopian Birr to USD. 

8. It would be interesting to deepen the discussion about if (and how) socio-cultural characteristics of the population could be related to the hesitancy to take the COVID-19 vaccine. In addition, insights on how to overcome this hesitancy should be proposed at the end of the discussion or in the conclusion.

 Thank you reviewer for important idea but identifying the association between socio-cultural characteristics and vaccine hesitancy was not objective of the study. Dear reviewer we have added the insights on how to overcome vaccine hesitancy in the conclusion.

9. In addition, some formatting errors, such as parentheses and capital letters, need to be corrected

 We have edited our manuscript in revised manuscript.

Reviewer # 2

1. It would be nice to have more data regarding the current situation in Ethiopia and the situation around the time of the study. Had cases continued to rise at that time? How bad was the area surveyed hit?

 Thank you reviewer for important observation. We have incorporated this comment with in the revised manuscript. You can see under introduction part starting from line 8.

2. Are there any data regarding public immunization willingness in Ethiopia? 

 Thank you dear reviewer for important insight. We have included the data regarding public willingness to COVID-19 vaccine in main manuscript in the introduction section. The findings regarding public willingness showed that willingness is low.

3. How well did the residents of Ethiopia accept public health recommendations for things like masks etc? This would help better frame the research.

 We have incorporated this comment with in revised manuscript. Dear reviewer you can get this in line 28 of introduction part.

4. Also- what is the current vaccine availability in Ethiopia? Are any vaccine available? 

 We have incorporated this with in revised manuscript. It is located in the last paragraph of the introduction part.

5. General: Recommend better editing for English throughout.

 Thank you dear reviewer for good recommendation. We have edited the revised manuscript.

6. COVID-19 should be capitalized throughout

 We have corrected it accordingly.

7. Abstract: Methods should be clear that this is a survey of healthcare workers at two hospitals.

 Tank you dear reviewer important insight. Our study is facility based cross-sectional study and it is not survey. The study was conducted in two hospitals and participants were selected randomly by lottery methods after proportional allocation to each hospitals based on their professional numbers. 

8. How was the survey conducted? Was it anonymous? Was it a paper survey or administered in person?

 Thank you dear reviewer for interesting questions. Data were collected by self-administered questionnaire. First participants were selected randomly by lottery method in both hospitals and data collectors provided questionaries’ to selected professionals. 

9. What types of healthcare workers were eligible? Beyond Nurses, what other types of healthcare workers were eligible to be included? Please also give more information on what "first degree holders" are.

 Thank you dear reviewer for important observations. Health care workers who involved in health care and direct contact with patients were eligible for the study. Beyond nurses all professionals including physician, midwifery, medical laboratory technologist, pharmacist, radiology technicians and psychiatry professionals were eligible and included within the study. First degree holder in our study is to mean those professional who have bachelors of Science degree (First degree) in any field of health like medicine, nursing, midwifery or others field.

10. Given that "Others" are the reference population, we need more information about that others" actually are. Consider making nurses or a different, larger, group the reference group. 

 Thank you dear reviewer for important insight. Others included radiology technicians and psychiatry professionals. We have changed the reference category based on your recommendation and in the revised manuscript we put nurses as a reference group.

Reviewer # 3

1. The authors are suggested to give more details on the context/ development of COVID-19 in (Southern) Ethiopia near the data collection period (March 2021), which could influence HCW’s acceptance of the vaccine. 

 Thank you dear reviewer for good observation. We have included the situation of COVID -19 in Ethiopia in the revised manuscript. It is located within the introduction part starting from line 7 of introduction part.

2. The scoring of COVID-19 attitude seemed problematic. I could not understand why the responses were ranked from positive (3 points), negative (2 points) to not sure (1 point). Should “not sure” score 2 points while negative 1 point? Furthermore, since there were 7 items, the total scores should range from 7 to 21, not 7 to 18. Can the authors show the internal consistency of these 7 items? In the Methods, please also make it clear that these measures are referring to attitude toward COVID-19 preventive measures.

 We appreciate dear reviewer for important observation. It was mistakenly written with in old version of the manuscript and your insight was correct. During analysis of the result, attitude responses was scored as follows (yes = 1, no = 0 and not sure = 0). And responses to reversed questions were reversed when assigning the points (Yes=0, not sure = 0 and No=1). As stated within the result, the attitude score ranged from 0 to 7. This scoring is consistent with previous study conducted in Healthcare Workers’ Attitude toward COVID-19. Dear reviewer this is link to access the previous study (https://doi.org/10.2147/JMDH.S287156). The mean attitude score (4.9 �1.6) was computed and participants' were considered as having a positive attitude if the attitude score ≥ mean attitude score. We have corrected the scoring of attitude items which was consistent with our result of attitude responses within revised manuscript. We have also included statements referring to attitude toward COVID-19 preventive measures in method parts.

 The internal consistency of 7 attitude items was 0.818 and we have uploaded the SPSS output of reliability in response letterer.

3. Responses to the question of COVID-19 vaccine hesitancy were dichotomized to “yes” and “no”. Can the authors presented the original scale of the responses?

 Thank you reviewer for important observation. We have evaluated self-reported vaccine hesitancy of the participants according to the WHO definition using previously adapted three questions. We have classified participants’ response to vaccine hesitancy to yes and no which is consistent to previous study (.DOI:https://doi.org/10.1016/j.jhin.2020.11.020).

4. The logistic regression models used in the study appeared to be “multivariable” (1 dependent variable, multiple independent variables) instead of “multivariate” (multiple dependent variables). Please correct. 

 Thank you for important comment. We have changed it in the revised manuscript.

5. In the abstract the sample size was 423, whereas in the main text it was 405. Please correct.

 We have corrected it with in the abstract part of the revised manuscript.

6. Table 2: Was there any particular reason to divide the chronic diseases into the two categories listed?

 Thank you dear reviewer for important observation. We categorized chronic disease that professionals’ had for description purpose. 

7. Table 4: Please indicates the correct answers for the questions on COVID-19 knowledge.

 Thank you for good review. We have presented the correct knowledge responses (bold) in the response letter.

8. Table 5: I have doubts on the question “Do you use non-conventional remedies (Honey, garlic, ginger, and lime) when you have flu-like symptoms?” as a measure of COVID-19 prevention practice. This dose not seem a proven/ recommended practice to prevent COVID-19.

 Good observation. These none-conventional remedies may increase the immunity of individuals in fighting the infection. The question is consistent with previous published literature (https://doi.org/10.2147/JMDH.S287156).

9. Please shows the overall prevalence with 95% CI of COVID-19 vaccine acceptance in the results and in the abstract.

 Thanks for important observation. We have included the 95% CI of COVID-19 vaccine acceptance in abstract and result.

10. Table 7: The percentages shown appeared to be cell percentages (rows and columns add up to 100%), which are not useful nor straightforward to interpret. The authors should use row percentages (percentages in the same row add up to 100%) to show the prevalence of vaccine acceptance in each subgroup of variables. Also, it seems more meaningful to use “Nurse” as the reference group instead of “Others” for the variable “Type of Profession”. Please also show the association of vaccine hesitancy and COVID-19 vaccine acceptance (not presented in the table) and which variables were adjusted in the adjusted models.

 Thank you dear reviewer for important observation. We have corrected the percentage of variables that’s sum to be give 100 %. We have also put nurse as reference category and association between vaccine hesitancy and COVID -19 vaccine acceptance. 

11. Discussion 3rd paragraph: It is surprising to see that vaccine hesitancy was not associated with COVID-19 vaccine acceptance, since many prior studies have shown vaccine hesitancy/ previous vaccination history to be strongly associated with vaccine acceptance. I could not interpret the results because they were not presented (see comment 10).

 We have corrected this in our revised manuscript. 

12. The vaccine acceptance among nurses appeared to rather low relative to other healthcare workers (~45% vs 54% based on the numbers presented in table 7). This is concerning because nurses constitute the largest health care workforce and have frequent contacts with patients. Can the authors discuss the finding?

 We are thankful for important observation. We have discussed this in the revised manuscript under discussion part.

13. The authors should elaborate the implications of their findings/ How their findings could be used to improve COVID-19 vaccine acceptance?

 We have incorporated this in the introduction and conclusion part of the revised manuscript.

14. The authors should discuss the limitations of their findings, which are absent in the article.

 We thanks for important recommendation. We have added this within revised manuscript. 

15. The article would be much benefited from English editing.

 We have corrected this in our revised manuscript.

---

## [Decision Letter · Decision Letter 1]

12 Aug 2021

PONE-D-21-13549R1

Health Care Workers Intention to Accept COVID-19 Vaccine and Associated Factors in Southwestern Ethiopia, 2021

PLOS ONE

Dear Dr. Angelo,

Thank you for submitting your manuscript to PLOS ONE. After careful consideration, we feel that it has merit but does not fully meet PLOS ONE’s publication criteria as it currently stands. Therefore, we invite you to submit a revised version of the manuscript that addresses the points raised during the review process.

We look forward to receiving your revised manuscript.

Kind regards,

Livia Melo Villar

Academic Editor

PLOS ONE

Journal Requirements:

Additional Editor Comments:

Dear Author,

Thanks for sending the revised manuscript, reviewers suggested minor revision what I also agree,

Best regards,

Livia

Reviewers' comments:

Reviewer's Responses to Questions

**Comments to the Author**

1. If the authors have adequately addressed your comments raised in a previous round of review and you feel that this manuscript is now acceptable for publication, you may indicate that here to bypass the “Comments to the Author” section, enter your conflict of interest statement in the “Confidential to Editor” section, and submit your "Accept" recommendation.

Reviewer #1: All comments have been addressed

Reviewer #2: (No Response)

Reviewer #3: All comments have been addressed

2. Is the manuscript technically sound, and do the data support the conclusions?

Reviewer #1: Yes

Reviewer #2: Yes

Reviewer #3: Yes

3. Has the statistical analysis been performed appropriately and rigorously? 

Reviewer #1: Yes

Reviewer #2: Yes

Reviewer #3: Yes

4. Have the authors made all data underlying the findings in their manuscript fully available?

Reviewer #1: Yes

Reviewer #2: No

Reviewer #3: Yes

5. Is the manuscript presented in an intelligible fashion and written in standard English?

Reviewer #1: Yes

Reviewer #2: Yes

Reviewer #3: Yes

6. Review Comments to the Author

Reviewer #1: Dear authors,

Most of my queries were addressed, so I would like to reccomend this manuscript for publication.

Reviewer #2: The authors claim it is not a two hospital survey but then the methods explicity state the study was conducted in MTUTH and GTSGH- two hteaching hospitals. They randomly selected these two hospitals out of a total of four in the southwestern part of ethiopia, but it still was only two hospitals. Thus, this is a fairly limited study given that it only explores attitudes in two teaching hospitals. This should be described as a limitation.

I'm confused still on how the sutvey was administered. "self-administered" isn't very descriptive. Was it a paper survey or on a computer? Were responses anonymous?

More detail on the type of healthcare workers should be included in the main manuscript.

Finally, there are a lot of tables. I suggest condensing these down.

Reviewer #3: The authors mostly addressed my comments. I had the following suggestions to improve the clarity of the paper further:

1. Please include the internal consistency results of the 7 items on attitudes toward COVID-19 preventive measures in the methods.

2. In the abstract, some results (e.g., AOR=15.18) seem not consistent with those presented in the table 7. Please also indicate the reference group for physicians.

3. Please indicate which item is reversed code in Table 4

7. PLOS authors have the option to publish the peer review history of their article (what does this mean?). If published, this will include your full peer review and any attached files.

Reviewer #1: No

Reviewer #2: No

Reviewer #3: **Yes: **Tzu Tsun Luk

---

## [Author Response · Author response to Decision Letter 1]

19 Aug 2021

 We have ensured that our reference list is complete and correct.

Comments to reviewers 

Reviewer # 2

1. The authors claim it is not a two hospital survey but then the methods explicity state the study was conducted in MTUTH and GTSGH- two hteaching hospitals. They randomly selected these two hospitals out of a total of four in the southwestern part of ethiopia, but it still was only two hospitals. Thus, this is a fairly limited study given that it only explores attitudes in two teaching hospitals. This should be described as a limitation.

 Thank you reviewer for this important observation. We have included this in limitation part of the revised manuscript. 

2. I'm confused still on how the sutvey was administered. "self-administered" isn't very descriptive. Was it a paper survey or on a computer? Were responses anonymous?

 Thank you reviewer for an interesting question. The study was paper survey and anonymous. The name of the participants was not included in the paper. No person identifier information was included in the paper. We have included this in Data Collection Tool, Quality Control, and Procedure and Ethical Considerations parts of the revised manuscript.

3. More detail on the type of healthcare workers should be included in the main manuscript.

 Thank you for important observation. We have added the types of health care worker included in the study in inclusion part of the revised manuscript.

4. Finally, there are a lot of tables. I suggest condensing these down

 Thank you for such important suggestion. But as journal requirement there is no limitation for numbers of table. And for more clarity of the finding we prefer to present by table as many of questions we used are long.

 Reviewer #3

1. Please include the internal consistency results of the 7 items on attitudes toward COVID-19 preventive measures in the methods.

 Thank you dear reviewer for important insight. We have included for this in the methods parts of the revised manuscript. 

2. In the abstract, some results (e.g., AOR=15.18) seem not consistent with those presented in the table 7. Please also indicate the reference group for physicians.

 We appreciate for such vision. We have corrected it in the abstract part of the revised manuscript. The reference group was nurses.

3. Please indicate which item is reversed code in Table 4.

 Thank you dear reviewer for such comment. We have indicated for reversed item during scoring in the table 4 of the revised manuscript by symbol *.

---

## [Editor Report · Decision Letter 2]

24 Aug 2021

Health Care Workers Intention to Accept COVID-19 Vaccine and Associated Factors in Southwestern Ethiopia, 2021

PONE-D-21-13549R2

Dear Dr. Angelo,

We’re pleased to inform you that your manuscript has been judged scientifically suitable for publication and will be formally accepted for publication once it meets all outstanding technical requirements.

Kind regards,

Livia Melo Villar

Academic Editor

PLOS ONE

Additional Editor Comments (optional):

Dear Author ,

Thanks for sending the revised version of this paper.

Sincerely

Livia Villar
---

## [Editor Report · Acceptance letter]

27 Aug 2021

PONE-D-21-13549R2 

Health Care Workers Intention to Accept COVID-19 Vaccine and Associated Factors in Southwestern Ethiopia, 2021 

Dear Dr. Angelo:

I'm pleased to inform you that your manuscript has been deemed suitable for publication in PLOS ONE. Congratulations! Your manuscript is now with our production department. 

Kind regards, 

on behalf of

Dr. Livia Melo Villar 

Academic Editor

PLOS ONE